# Rock Wool Fiber-Reinforced and Recycled Concrete Aggregate-Imbued Hot Asphalt Mixtures: Design and Moisture Susceptibility Evaluation

Farah Khaleel Hussein [1], Mohammed Qadir Ismael [1,*] and Ghasan Fahim Huseien [2,*]

1 Department of Civil Engineering, University of Baghdad, Baghdad 10071, Iraq; farah.hussein2001m@coeng.uobaghdad.edu.iq
2 Department of the Built Environment, College of Design and Engineering, National University of Singapore, Lower Kent Ridge, Singapore 117566, Singapore
* Correspondence: drmohammedismael@coeng.uobaghdad.edu.iq (M.Q.I.); ghasanfahim@gmail.com (G.F.H.); Tel.: +65-83057143 (M.Q.I. & G.F.H.)

**Abstract:** Designing asphalt mixtures for pavement construction by controlling the moisture-mediated damage remains challenging. With the progression of time, this type of damage can accelerate deterioration via fatigue cracking and rutting unless inhibited. In this study, two types of hot asphalt mixtures (HAMs) were made by incorporating recycled concrete aggregates (RCAs), which were reinforced with rock wool fibers (RWFs). The first specimen was a normal mixture with a completely virgin aggregate, and the second one was a sustainable mixture with 30% RCAs. The proposed mixes were thoroughly characterized to assess the impact of RWF incorporation at various contents (0.5, 1, 1.5, and 2%) on moisture resistance. The optimal asphalt concentration (OAC) and volumetric parameters of the mixes were determined using the Marshall technique. The moisture susceptibility of the obtained HAMs was evaluated in terms of the tensile strength ratio (TSR). The results revealed that the moisture resistance, Marshall stability, flow, and volumetric parameters of the HAMs were improved due to the reinforcement by RWFs, indicating a reduction in the moisture sensitivity and an increase in TSR%. In addition, the HAMs designed with 1.5% RWFs displayed the highest TSR% (11.37) and Marshall stability compared to the control mix. The observed improvement in the moisture resistance and Marshall attributes of the prepared HAMs was ascribed to the uniform distribution of the RWFs that caused a well-interconnected structure and tightening in the asphalt concrete matrix. It is asserted that the proposed HAMs can be nominated for the construction of durable high-performance pavements.

**Keywords:** moisture susceptibility; hot asphalt mixtures; rock wool fibers; recycled concrete aggregates; tensile strength ratio

## 1. Introduction

Regarding socio-economic growth, the importance of highways is unquestionable, wherein transportation systems remain a key sector. In recent years, with rapid changes in socio-economic and cultural affairs in Iraq, the number of road vehicles has been substantially increased. Consequently, Iraqi highways have been receiving immense loads due to there being no alternatives within the transportation sector for people and commodities to utilize for travel [1]. In Iraq, highways are often adversely affected by various factors because they are mostly made of asphalt concrete pavement. Moisture-related damage is the most significant factor that notably reduces the structural capacity of asphalt pavement, thus making it somewhat unusable. To overcome this problem, highway engineers have been trying to improve asphalt concrete pavements' resistance toward moisture by incorporating several types of additives [2].

The performance of a road lies in its ability to meet heavy traffic flow and environmental demands over its design life [3]. It is well known that environmental factors and traffic loads can rapidly deteriorate road pavements; thus, durable high-performance concrete pavements for highways are required. According to Sebaaly [4], hot asphalt mixture (HAM) performance is essentially determined by its resistance to fatigue, rutting, raveling, and low-temperature-mediated cracking. HAMs' resistance against these types of damage can be determined using performance measurements that involve sensitivity to moisture and temperature. Thus, evaluation of the moisture susceptibility of HAMs is of high significance because the aforementioned processes can lead to various types of damage that can occur either separately or simultaneously. The result of moisture-mediated damage in HAMs can lower their overall stiffness or strength, thus weakening durability performance. Consequently, any of the failure modes like raveling, low-temperature cracking, and rutting can occur if a HAM is susceptible to moisture-related damage [4,5].

The efficiency of HAM-activated pavements can be affected by moisture sensitivity which essentially loosens adhesion bonds amid the aggregates' surface and asphalt binder and/or weakens the cohesion inside the binders or aggregates. Various factors including the physicochemical qualities of the binders and aggregates, the design methods, and the construction features can appreciably contribute to the complication of HAM-made pavement's susceptibility toward moisture. When moisture-mediated damage occurs in HAM-based pavements, it can cause severe worsening within asphalt network structures (like bleeding, cracks and ruts, untangling, and local failure). These types of damage are similar to those caused by other issues related to materials' qualities, designs, and manufacturing processes. Therefore, it becomes essential to determine whether pavement damage is caused by moisture or is due to other factors, thus requiring an appropriate solution as a remedial measure [6]. Various laboratory tests can be used to examine the moisture sensitivity of HAMs by treating them with anti-stripping additives [7]. Currently, fibers, as a class of additives, have received much attention due to their distinct properties. Different types of fibers have been applied in HAMs to improve their moisture resistance, as network structure modifications have been attributed to the enhanced performance of HAM-based pavements [8].

Several studies [9–12] have reported that fibers are a class of additives that can be used as effective reinforcement materials in HAMs to enhance their moisture resistance. Many types of modifying fibers, such as polyester, cellulose, and minerals, are extensively used in diverse asphalt mixes to improve their resistance against moisture [13]. It has been acknowledged that by raising the fiber contents to a critical value in HAMs, moisture resistance can be enhanced; however, this depends on the fibers' nature and span [14]. Essentially, fiber reinforcement can increase the tensile characteristics of HAMs, thus preventing crack formation due to excessive mechanical load, shrinkage, and temperature variations [15,16]. Fibers used to reinforce asphalt mixture pavements were found to improve service life, stiffness, and strength performance [17]. Based on these factors, this work intends to investigate the effect of rock wool fiber (RWF) reinforcement on the Marshall traits, volumetric parameters, and moisture susceptibility of recycled concrete aggregates (RCAs) infused in HAMs used for highway pavement construction.

Numerous studies [15,18,19] have explored which components and/or modifications improve the properties of hot mix asphalt and reduce or completely prevent the formation of asphalt pavement distress. Accordingly, various types of fibers have been employed in asphalt pavements to increase performance [20]. These fibers have improved asphalt pavement stiffness and strength, thus allowing structures to transfer loads effectively into pavement mixtures. Likewise, the addition of fibers to asphalt was shown to enhance mixes' characteristics, thereby promoting sustainability [21]. This, in turn, was shown to improve the material's service life, requiring less frequent maintenance [11]. In addition, some studies demonstrated that fiber-reinforced asphalt concrete can obtain strong resistance against aging, moisture damage, bleeding, and fatigue and reflection cracking [11,22]. Generally, reinforcement implies the addition of materials with certain desirable properties

into another with other attributes. Fiber reinforcement is performed to increase a material's ability to carry tensile loads, prevent distress development, and stop crack formation and spread [23]. As a result, asphalt mixtures with sufficient fiber contents are stronger and more resistant to permanent deformation [16].

The recycling of various industrial and agricultural wastes is a relatively new strategy for transforming them into new products, thus reducing waste-mediated environmental pollution and landfill problems. Furthermore, this recycling can enhance the production of low-cost sustainable construction materials from fresh natural sources [24]. Concrete is a major construction component and source of demolition wastes. Worldwide, the estimated annual production of building and demolition waste is 1183 million tons [25]. For many years, the use of concrete demolition wastes as recycled aggregates to replace natural aggregates has generated much interest among researches in construction industries. Several studies [26,27] reported that the use of concrete demolition wastes as recycled aggregates in concrete can be beneficial for both the environment and profits for the concrete industry. Annually, nearly 144 thousand tons of solid wastes are produced in Iraq, of which about 68% of it originates from buildings and demolitions [28]. However, RCAs are far from being utilized in the production of new concrete, stabilized soil, and road pavements [29]. In comparison to natural aggregates, RCA particles are weaker, can absorb more water, have a lower density, and are less resistant to abrasion. All of these demerits of RCA must be surmounted before it is utilized in construction sectors.

Due to the lack of landfills and high cost of transportation, waste management poses a severe environmental concern. As a result, RCAs have emerged as a sustainable alternative to asphalt mixtures. It is known that RCAs can be used to stabilize soil and make new concrete as well as road pavements, especially for unbound and sub-base layers [29]. At present, more than 85% of RCAs are used as road bases in the USA [30]. The main benefits of potential RCA uses depend on sustainable growth principles such as lowering waste production and its adverse impacts on the environment, natural resource conservation, and cost reduction of demolished concrete waste disposal [31]. Paranavithana and Mohajerani [32] investigated the effect of RCAs on the characteristics of asphalt concrete wherein 50% RCA was used as coarse aggregate. The addition of RCAs into HAM was shown to decrease the bulk density, voids in the mineral aggregates, and asphalt-filled voids, thus improving the elastic modulus, asphalt coating thickness, and stripping ability. According to Topal et al. [33], RCAs can be added in place of HAM aggregates to achieve desired Marshall stability and ITS of mixes. With increasing levels of RCAs in various mixes, Marshall traits were improved. Both mineral aggregates' voids and asphalt-filled voids (VFA) were decreased, which was ascribed to the Marshall compactor crushing of the RCAs during the compaction process. The volumetric and mechanical properties of mixes made using RCAs were found to be better compared to conventional mixes [34]. Mills-Beale and You [35] examined the behavior of hot mix asphalt containing RCAs using the super-pave mix design method. Both VMA and VFA were reduced with an increase in RCA contents in the mixes [36].

Considering the importance of novel sustainable HAMs for highway pavement construction, we designed RWF-reinforced and RCA-imbued HAMs. The obtained samples were characterized to determine the influence of various RWF contents on the moisture susceptibility of the proposed HAMs. The moisture resistance and Marshall properties of the obtained HAMs during a wearing course were tested and compared with a control mix. The moisture susceptibility assessment was performed in terms of local parameters like the index of retained strength (IRS) and global parameters such as the tensile strength ratio (TSR).

## 2. Materials and Methods

This study was conducted in five phases. The first phase involved the preparation of raw materials such as asphalt binder, coarse and fine aggregates, mineral fillers, and rock wool fibers. The second phase dealt with the synthesis of RCAs with the desired sieve size. In the third phase, asphalt cement mixtures were made to determine the OAC

for the control mixture. In the fourth phase, the OAC values of the specimens containing different percentages of RWF were determined. Next, the OAC values of the mixture were obtained depending on the Marshall properties. Finally, the specimens' Marshall stability and flow values were determined together with the volumetric properties to comply with the requirements of Iraqi standards. The values of TSR were obtained using the ITS test.

### 2.1. Materials Characterization

Locally available materials were used in this study. Presently, these materials are used in Iraq to construct roads and highways. As basic constituents of novel HAMs, RWFs were purchased from the local market. Asphalt cement made by the Al-Daurah Refinery with a penetration grade of 40–50 was procured (Table 1), meeting the standards of SCRB (R/9, 2003) [37]. Crushed coarse aggregates with a nominal maximum size of 1/2 inch (12.5 mm) were collected from the hot mix plant of Al-Nibaee quarry (Baghdad Mayoralty) as per SCRB standards (R/9, 2003). Fine aggregates (river and crushed sand) were collected from the same source with particle sizes ranging from 4.75 mm (passed through sieve No. 4) to 0.075 mm (retained on sieve No. 200) as per SCRB standards (R/9, 2003). The mineral fillers in the asphalt mixture were the part of the aggregates that passed through sieve No. 200 (0.075 mm). Dried limestone dust was utilized as filler in the designed asphalt mixes.

**Table 1.** Physical properties of the rock wool.

| Properties/Materials | | Description | | | | | | | Tolerance | Standard |
|---|---|---|---|---|---|---|---|---|---|---|
| | | Stone Wool (SL1) | | | | Stone Wool (SL2) | | | | |
| Density ($\gamma$), kg/m$^3$ | | 70 | | | | 110 | | | ±10% | [38] TS EN 14303 |
| Width (w), mm | | 600 | | | | | | | ±1.5% | [39] TS EN 822 |
| Length (l), mm | | 1200 | | | | | | | ±2% | [39] TS EN 822 |
| Thickness (t), mm | | 25 | 40 | 50 | 60 | 80 | 100 | 120 | T4 | [40] TS EN 823 |
| Facing | | Un-faced | | | | | | | - | |
| Reaction to fire | | A1 | | | | | | | - | [41] TS EN 1350-1 |
| Declared thermal conductivity ($\lambda_o$), W/m.K | T, °C | 50 | 100 | 150 | 200 | 250 | 300 | 350 | - | [42–44] TS EN 1266, 12939 and 13787 |
| | SL1 | 0.039 | 0.048 | 0.059 | 0.072 | 0.087 | 0.103 | 0.122 | - | |
| | SL2 | 0.037 | 0.045 | 0.053 | 0.062 | 0.073 | 0.085 | 0.097 | - | |
| Squareness (So), mm/m | | Max. 5 | | | | | | | - | [45] TS EN 824 |
| Water vapor diffusion resistance coefficient ($\mu$) | | 1 | | | | | | | - | [46] TS EN 12086 |
| Packaging material | | PE Film | | | | | | | - | - |
| Other information | | Yellow–black glass tissue/aluminum foil-faced types are also available. | | | | | | | - | - |

A demolished residential building (Haifa Street, Baghdad constructed in 1980) was used as a source of RCAs to design the HAMs. First, concrete pieces were crushed by a Los Angeles machine to remove the mortar around the gravel particles (Figure 1). Next, the coarse particles were immersed in $CH_3COOH$ acid (5% of molarity) for 24 h. After immersion, the recycled concrete aggregates were washed thoroughly using water and dried until a dry status of the saturated surface was reached. Later, they were sieved to obtain the required granular distribution based on SCRB standards (R/9, 2003). The designed HAM contained 30% RCAs by weight of the total aggregates as coarse aggregates, implying 73% by weight of coarse aggregates. After mechanical sieving of the coarse and fine aggregates, they were separated into groups according to particle size. Gradation with a nominal maximum size of 12.5 mm was used. The blending percentage for the Type III-A wearing coarse was chosen as per SCRB stipulation (2003) (Figure 2).

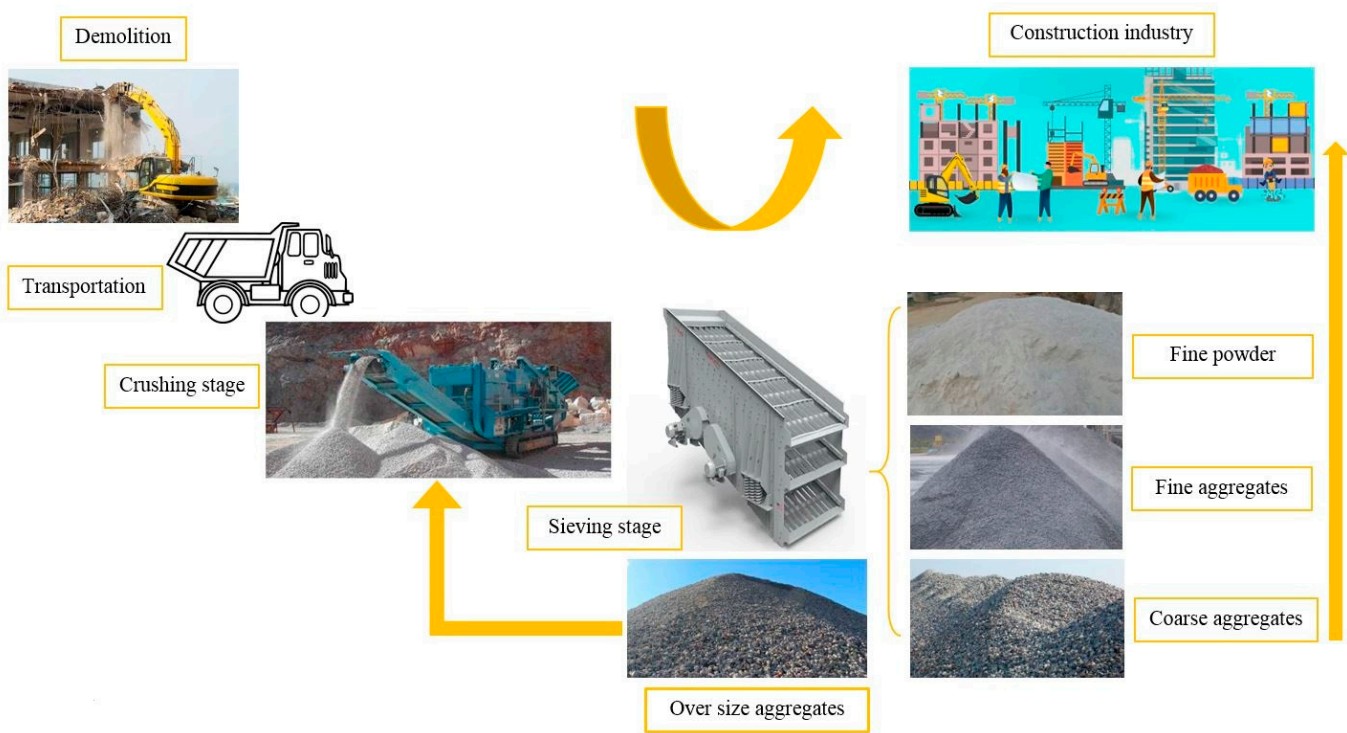

**Figure 1.** Recycling process of the demolished concrete.

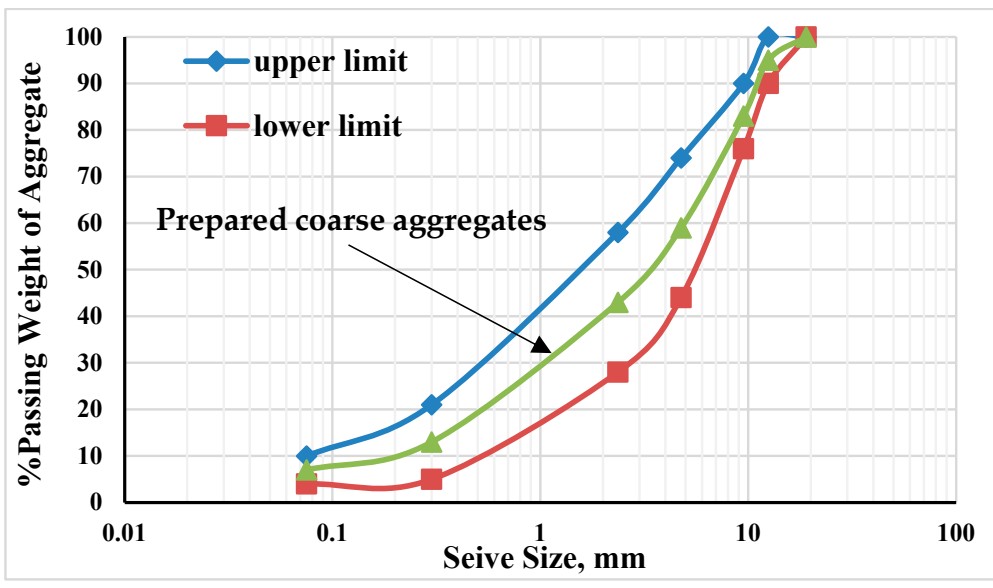

**Figure 2.** Specification limits and mid-point gradation of coarse aggregates.

Basalt is a primary component of rock wool and was transformed into fibers after heating at 1350–1400 °C, thus producing RWF. However, it can be made in different sizes based on the intended purpose [47]. Table 1 shows the obtained tock wool (IZOCAM, Turkish company) and its physical properties. Figure 3 displays the appearance of the RWF and Figure 4 shows its SEM images. Fibers were added to the asphalt mixtures using the dry method, wherein the fibers were uniformly dispersed into the network structures [48,49]. In this work, the RWFs were fully dispersed into the aggregates to achieve the best mixing.

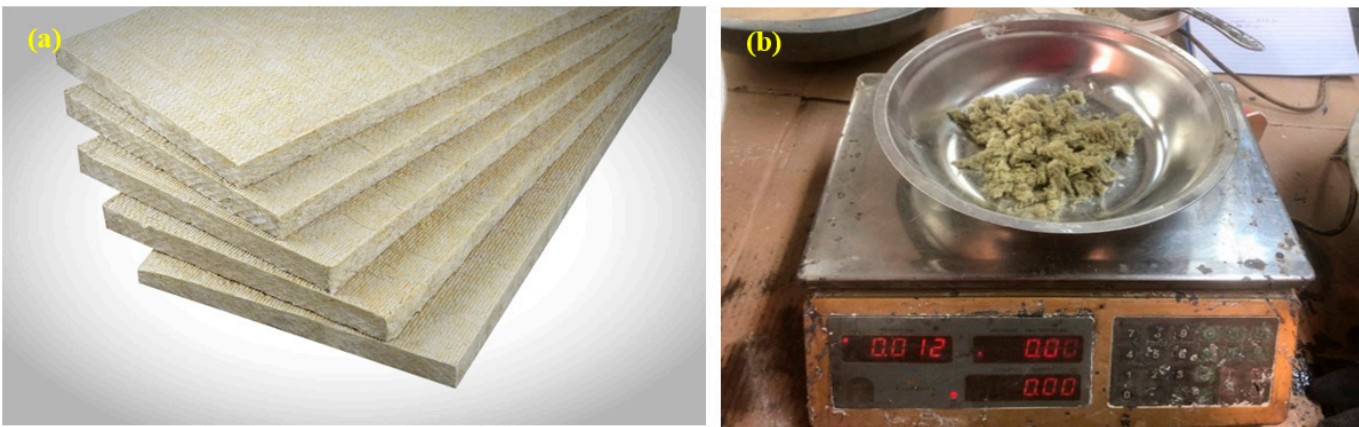

**Figure 3.** Physical appearance and SEM images of the RWF (**a**) before cut and (**b**) after cut.

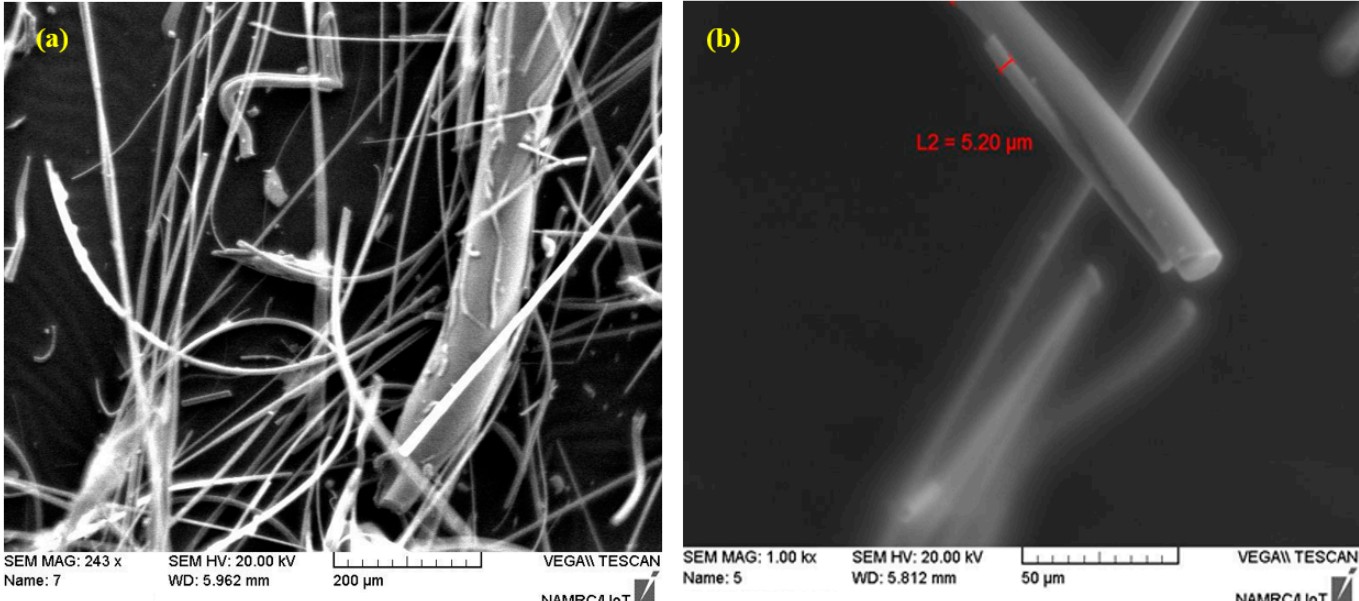

**Figure 4.** Physical appearance and SEM images of the RWF: (**a**) 200 μm and (**b**) 50 μm.

### 2.2. Mix Design

In the design process of the specimens, the aggregates were first dried at 110 °C to a constant weight and then sieved to obtain the required sizes. Next, they were blended with mineral fillers to obtain the desired gradation following the SCRB standard [37]. Using a controlled furnace, the coarse and fine aggregates were combined with mineral fillers of specified amounts and heated up to 170 °C. Simultaneously, the asphalt cement was heated to produce a kinematic viscosity of (170 ± 20) Centistokes. Then, it was weighed to the required amount based on the obtained OAC. Afterward, the created mixture of the coarse aggregates, fine aggregates, and mineral fillers was blended on a hot plate for approximately 3 to 4 min until the entire mixture containing aggregate particles was coated with asphalt.

### 2.3. Marshall Test

Following the ASTM D2726-08 standard [50], cylindrical specimens with a corresponding length and diameter of 63.5 mm and 101.6 mm were made for Marshall stability and flow measurements at 60 °C at a 50.8 mm/min load rate. This method used the Marshall apparatus to assess the hot asphalt mixtures' resistance against the plastic flow of the cylindrical specimens loaded on the lateral surface according to the ASTM D 6927-04

standard [51]. This technique included the preparation of cylindrical specimens of diameter 4 inches (101.6 mm) and of height 2.5 ± 0.05 inches (63.5 ± 1.27 mm). A hot plate containing a spatula, Marshall mold, and compaction hammer was heated at 150 °C. The preheated mold was filled with the asphalt mixture. Then, it was firmly spaded with the heated spatula 10 times inside the specimen and 15 times around the perimeter. The temperature of the mixture immediately after compaction was in the range of 142–146 °C. The compaction hammer having a sliding weight of 4.535 kg was used to apply 75 blows to the top and bottom of the specimen by a free fall from 18 inches (457.2 mm) of altitude [52]. The specimen was taken out of the mold after it was cooled inside the mold for 24 h at room temperature.

The Marshall flow and stability for each specimen were measured. First, the cylindrical specimens were submerged into hot water (60 °C) for about 30 to 40 min. Next, they were compressed along the lateral surface at a load rate of 2 inch/min (50.8 mm/min) until the highest load (failure) was obtained, determining the highest load resistance and flow values. For each group, a total of 3 mixes were made to obtain the mean value. For every specimen, the values of the bulk and highest (theoretical) specific gravity, density, and AV% were estimated following the relevant standards (ASTM D-2726-08 [51], ASTM D-2041-03 [53], and ASTM D-3203-05 [54]). The OAC values were chosen as the average content for maximum stability, 4% of air voids, and maximum bulk density following the SCRB standard (R/9, 2003).

### 2.4. Tensile Strength Ratio Test

The TSR test was conducted to evaluate the effect of moisture on the proposed HAMs made without and with RCAs. For this test, five trial additive-free specimens (conforming to the job-mix formula) were made with a diameter and length of 63.5 mm inch and 101.6 mm, respectively. Later, these specimens were subjected to different numbers of blows (45, 55, 65 and 75) to obtain the actual number that could achieve a target air void content of 7 ± 1%. Fifty-one blows were used to prepare six specimens of the asphalt mixture. These obtained specimens were divided into two groups, each with three specimens. The first subset (unconditional specimens) was immersed in water (25 °C for 30 min) to measure the ITS. The second group (conditional one) was tested with a certain process as mentioned in ASTM D-4867-09 [55]. The specific gravity, height, and AV% of the specimens were measured after placing both groups into the loading apparatus where the strips were positioned parallel and centered on the vertical diametric plane of the specimen. A diametric load at a rate of 50 mm/min (2 in/min) was applied until the maximum load was reached, after which the failure load was recorded (Figure 5). The ITS values of the specimens were calculated using the following equation:

$$S = (2000\,P)/(\pi\,t\,D) \tag{1}$$

where ITS is the tensile strength (kPa), P is the maximum load (N), t is the height immediately before the tensile test (mm), and D is the diameter (mm) of the specimen.

The tensile strength ratio (TSR%) was calculated via the following equation:

$$TSR = ((ITS\ con.)/(ITS\ uncon.)) \times 100 \tag{2}$$

where $ITS_{con}$ and $ITS_{uncon}$ are the average TS values of the moisture-conditioned and dry subset (kPa), respectively.

### 2.5. Index of Retained Strength Test

The IRS test was conducted to measure compressive strength (CS) reductions occurring due to moisture damage to the compacted asphalt mixtures containing binder. The CS values of the duplicated specimens were compared with those specimens submerged in water under the specified conditions. This enabled us to determine the development of a numerical index of the reduced CS. Essentially, this test was an indicator of the bitumen–aggregate mixes' moisture sensitivity as per the ASTM D 1075 standard [56]. A total of

6 cylindrical specimens with a length and diameter of 101.6 mm and 101.6 mm, respectively, were prepared following the procedure described in ASTM D 1074 [57]. Using this protocol, a trial mix was first prepared to determine the proper weight of the components which was about 1900 gm.

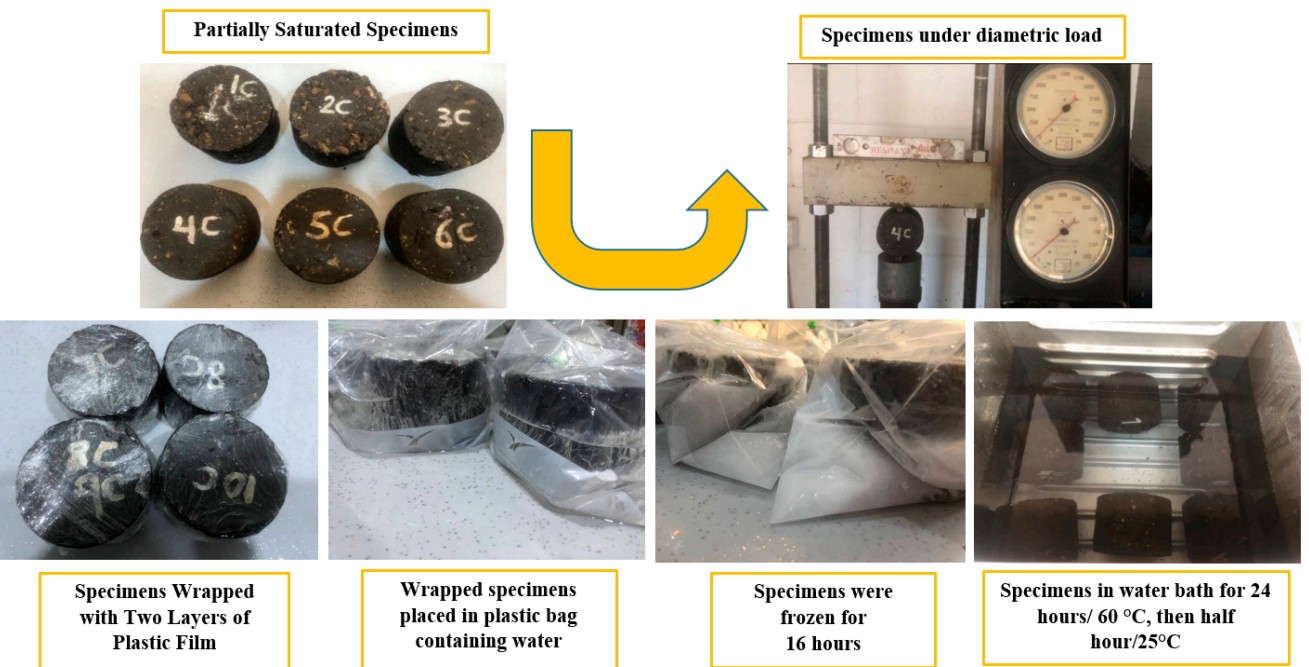

**Figure 5.** Specimen under ITS test.

For the Marshall stability test, all specimens were preheated in an oven followed by a rapid mixing of the heated aggregates with the desired weight of the asphalt cement. Then, after pouring half of the asphalt mixture into the mold, it was vigorously spaded with a heated spatula 15 times around the edge of the mold to prevent honeycombing and 10 times randomly over the asphalt mixture. The remaining half was quickly moved to the mold cylinder and spaded in the same way. Under an initial force of 1 MPa (150 psi), the mixture was pressed at the top and bottom to compress it up against the sides of the mold. Next, the specimen was subjected to the required load of 3000 psi (40,000 lb) for 2 min followed by room-temperature cooling for 24 h.

In accordance with ASTM D 2726 [50] procedures, the bulk specific gravity of the specimens was measured after ejecting them from the molds. The 6 tested specimens were divided into two groups wherein each group consisted of 3 mixes, ensuring the same mean bulk specific gravity for each group. The first group of the specimens were examined under dry conditions by placing them in an air bath (25 °C) for 4 h. Next, an axial load (5.08 mm/min or 0.2 in/min) was applied until failure occurred (the recorded failure load was denoted as S1). The second set of specimens were placed in a water bath (60 °C) for a day before being moved to another bath (25 °C) for 2 h, reaching the testing temperature. Then, the specimens were subjected to the same load rate (the recorded failure load was denoted as S2). The values of the IRS for various steps were calculated using the following equation:

$$\text{Index of retained strength} = [S2/S1] \times 100 \qquad (3)$$

(minimum 70% according to SCRB-R/9, 2003)

where S1 and S2 are the CS values of the dry group 1 (dry) and group 2 (wet) specimens, respectively.

## 3. Results and Discussion

### 3.1. Marshall Stability and Volumetric Properties

The Marshall stability, flow, and volumetric parameters of the control mix and HAMs made with various RWF contents and 30% RCAs are presented in Table 2. Figure 6 illustrates the RWF content-dependent variation in the stability of the mixes. The results revealed that the HAM prepared with RWFs (without RCAs) had greater stability than the control mixture. The stability of the HAM was increased from 16.16 to 55.56% with a corresponding increase in the RWF content from 0.5 to 2%. The HAM made with 1.5% RWF demonstrated the greater stability (Figure 6). This improvement in the stability of the HAM was mainly due to the uniform distribution of RWFs in the mixture, creating a 3D network structure. Consequently, the proposed HAM became highly resistant to shear flow and strongly bonded via the connecting points among the aggregate grains with enhanced adhesion. The observed decrease in the Marshall stability of the HAM containing 2% RWF was anticipated because at high densities, the fibers are long enough to weaken the network bonding in the mix. In addition, at a high RWF content, the distribution of the fibers in the HAM network might not be uniform, thus lowering the Marshall stability. The HAM containing 30% RCA showed greater stability compared to control specimen. Figure 6 clearly indicates that the stability of the HAM made with RCAs was significantly lower than those containing only natural aggregates. This was due to the presence of rounded aggregates used in the cement concrete mixtures. The interlocking of the rounded aggregates in the mixtures was much weaker than the interlocking of the crushed aggregates, indicating the proper implementation of the crushed aggregates in the asphalt concrete. It was shown that [36] RCAs had a significant effect on the HAM characteristics. It was asserted that the Marshall stability and flow as well as the volumetric properties of the proposed asphalt mixtures satisfied the necessary technical standards.

**Table 2.** Results obtained using the Marshall test.

| RWF (%) | Asphalt Content (%) | Marshall Properties | | | Density and Voids | | | |
|---|---|---|---|---|---|---|---|---|
| | | Stability (kN) | Flow (mm) | Bulk Density (gm/cm$^3$) | AV (%) | VMA (%) | VFA (%) |
| 0% of RCA | | | | | | | | |
| 0 | 4.9 | 9.9 | 3.5 | 2.32 | 3.50 | 15.89 | 77.98 |
| 0.5 | 5 | 11.5 | 3.39 | 2.3 | 4.29 | 16.70 | 74.33 |
| 1 | 5.05 | 13.2 | 3.28 | 2.297 | 4.65 | 16.85 | 72.41 |
| 1.5 | 5.1 | 15.8 | 3.32 | 2.293 | 4.82 | 17.04 | 71.74 |
| 2 | 5.3 | 15.4 | 3.4 | 2.281 | 5.31 | 17.65 | 69.89 |
| 30% of RCA | | | | | | | | |
| 0 | 5.1 | 9.7 | 3.6 | 2.304 | 4.36 | 16.64 | 73.81 |
| 0.5 | 5.2 | 10.9 | 3.45 | 2.298 | 4.61 | 16.95 | 72.81 |
| 1 | 5.3 | 13.8 | 3.41 | 2.29 | 4.94 | 17.32 | 71.48 |
| 1.5 | 5.55 | 13.5 | 4.2 | 2.285 | 5.15 | 17.72 | 70.95 |
| 2 | 5.7 | 13.2 | 5.2 | 2.2804 | 5.34 | 18.02 | 70.37 |

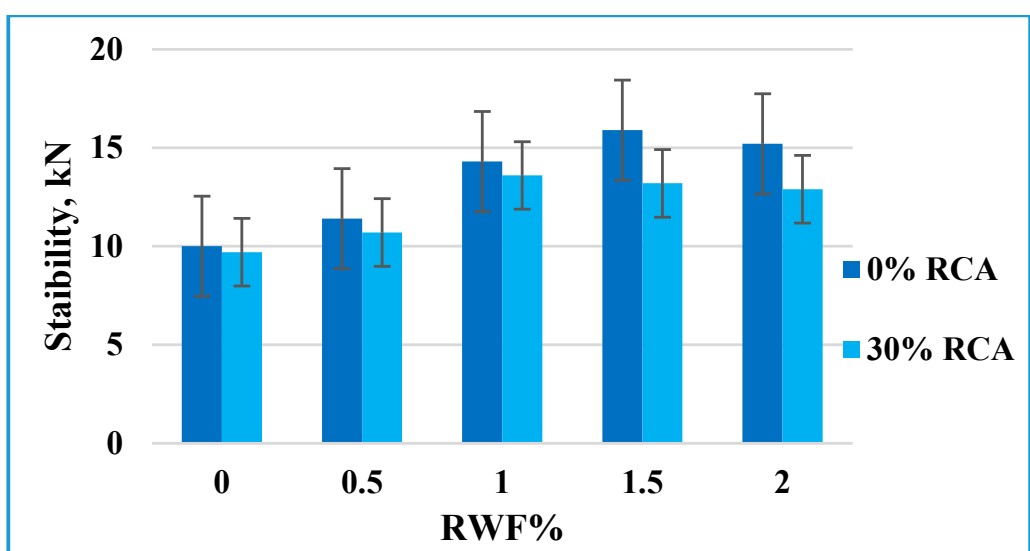

**Figure 6.** RWF content-dependent variation in the stability of the mixes.

Figure 7 shows the RWF content-dependent variation in the flow of the mixes. The observed high flow values of the prepared HAMs could result from the low resistance of the asphalt mixes against the applied loads. For all additive percentages, the flow value was smaller than that of the control mix. The obtained values of the Marshall flow at 0% RCA were 3.14, 6.29, 5.14, and 2.86 for the corresponding RWF contents of 0.5, 1, 1.5, and 2%. The flow value of the mixture made with 1% RWF and 30% RCA was equal to the flow of the conventional control mix. In comparison to the control mix, the newly designed sustainable HAMs showed lower flow values of 4.32, 5.41, 8.92, and 9.46% for the corresponding RWF contents of 0.5, 1, 1.5, and 2% (at 30% RCA). In short, the flow values of the HAMs declined as the RWF contents increased, indicating an increase in the stiffness of the fibers that made the mixes' network structure less flexible. Nevertheless, the measured flow of the HAMs was within the specified range of 2 to 4 mm. The obtained drop in the flow values with RWF reinforcement was in good agreement with other findings [58].

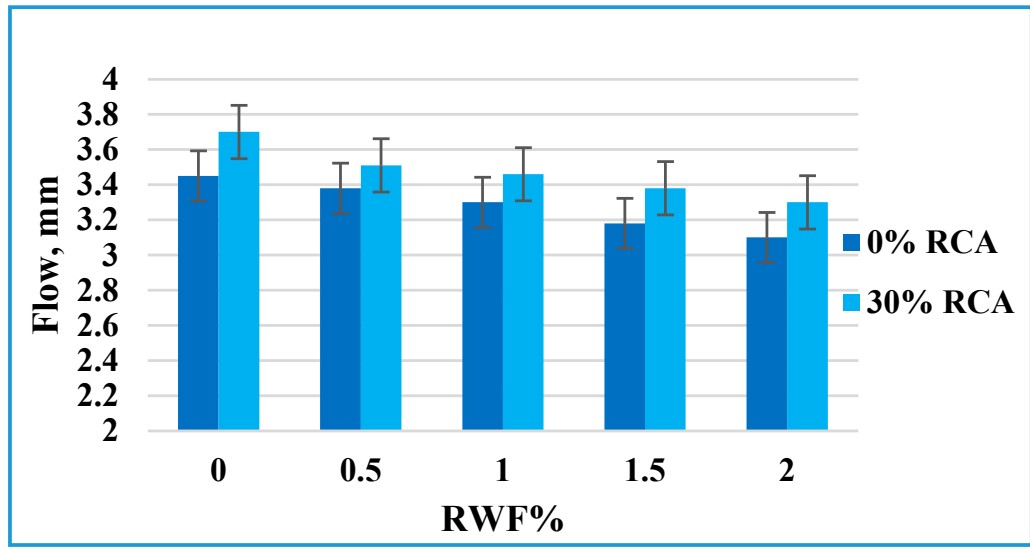

**Figure 7.** RWF content-dependent variation in the flow of the mixes.

Figure 8 illustrates the RWF content-dependent variation in the bulk density of the mixes. The bulk density values of the HAMs were 0.86, 0.99, 1.16, and 1.68% for the corresponding RWF contents of 0.5, 1, 1.5, and 2%. Compared to the sustainable control

mix (0% of RCA), the HAM made with 30% RCA showed a lowering in the bulk density from 0.26 to 1.02% when the RWF content was increased from 0.5 to 2%. This behavior indicated that the mixtures became harder to compress as the fiber content was increased, requiring more compactness to attain a higher density because the density of RWFs was much lower compared to the asphalt and aggregates.

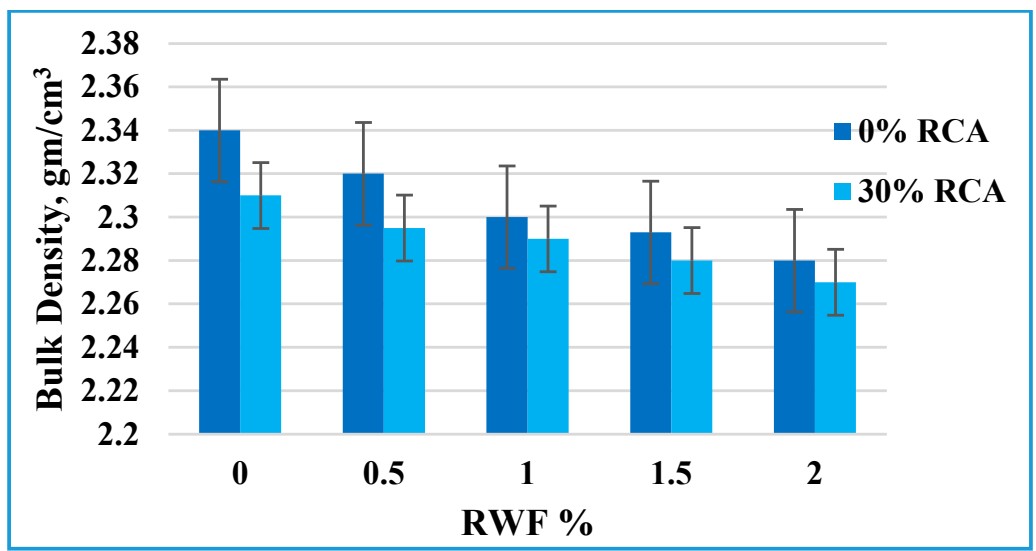

**Figure 8.** RWF content-dependent variation in the bulk density of the mixes.

Figure 9 displays the RWF content-dependent variation in the air voids (AV) of the HAM. The values of AV% were increased by 22.53, 32.90, 37.65, and 51.89% with corresponding increases in RWF contents of 0.5, 1, 1.5, and 2%. Compared to the sustainable control mix, the AV values of the RWF-reinforced specimens (at 30% RCA) showed higher AV% values of 5.71, 13.33, 18.10, and 22.48% for the corresponding RWF contents of 0.5, 1, 1.5, and 2%. In addition, the values of VFA% provided information regarding the holes between the aggregates in the compacted mixture filled with asphalt binder. VFA aimed to provide excellent durability to HAMs created from thin asphalt films on aggregates [59].

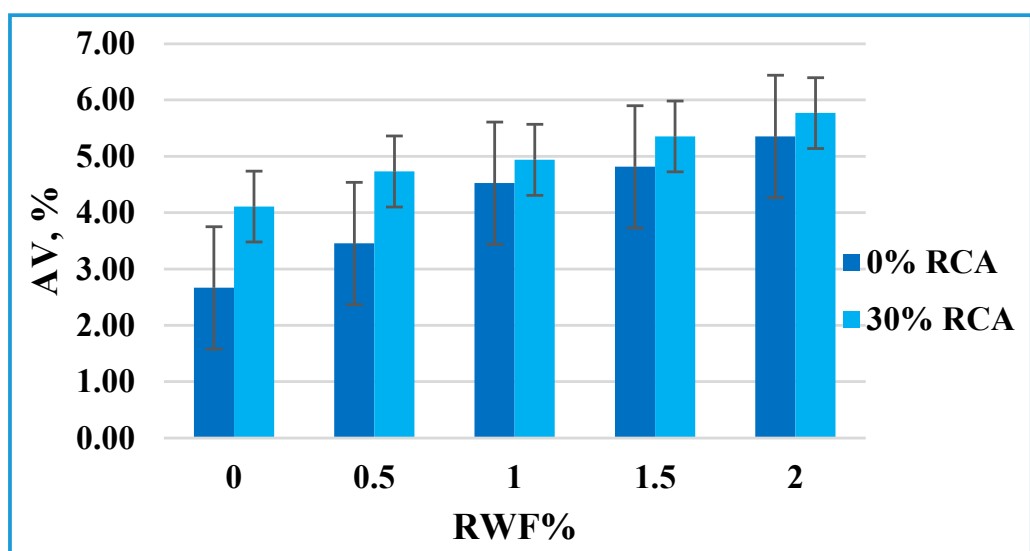

**Figure 9.** RWF content-dependent variation in the air voids of the mixes.

Figure 10 displays the RWF content-dependent variation in the VFA% of the mixes. The values of VFA% for the designed HAMs were reduced by 4.68, 7.14, 8.00, and 10.37 with corresponding increases in RWF contents of 0.5, 1, 1.5, and 2% RWF (at 0% of RCA). Conversely, the HAMs containing 0, 0.5, 1, 1.5, and 2% RWF (with 30% of RCA) showed a corresponding decrease in the VFA% of 1.35, 3.15, 3.87, and 4.66% compared to the sustainable control mix. This observation can be ascribed to the fibers clustering in the HAM network that prevented the asphalt from filling the entire space due to the absorption by RWF [37].

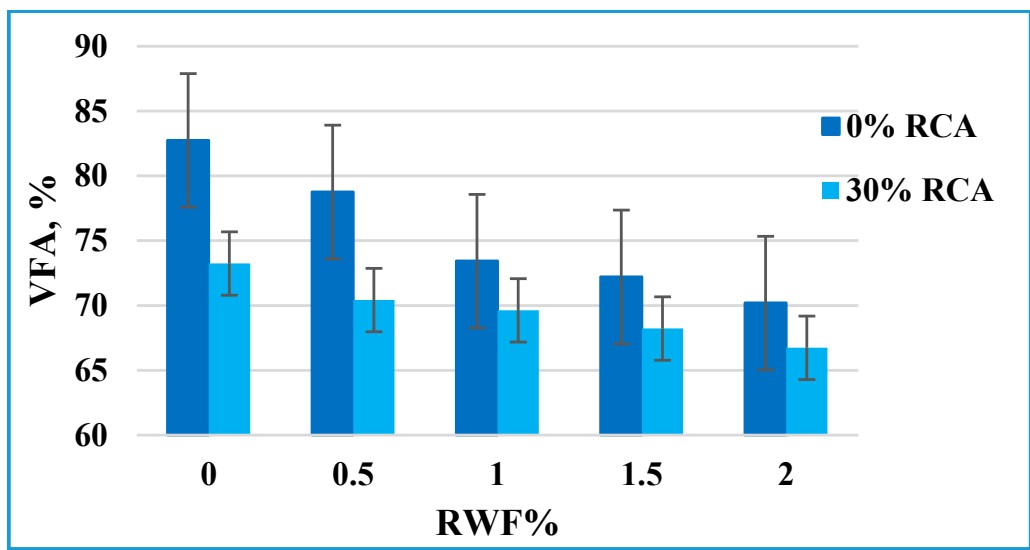

**Figure 10.** RWF content-dependent variation in the VFA of the mixes.

Figure 11 shows the RWF content-dependent variation in the VMA of the mixes. The percentage of voids in the mineral aggregates (VMA%) includes AV and voids filled with the effective asphalt. The VMA% must be sufficiently high to achieve a high-performance mixture. Low VMA% is detrimental to the durability due to the absence of sufficient amounts of bitumen to coat the aggregates. The results showed that compared to the control mix, the RWF-reinforced HAM made with 0% of RCA produced higher VMA% values of 5.25, 10.20, 12.12, and 16.27% at RWF contents of 0.5, 1, 1.5, and 2%, respectively. The sustainable HAMs containing 30% RCA showed VMA% of 4.15, 5.92, 9.73, and 12.96% for the corresponding RWF contents of 0.5, 1, 1.5, and 2%. The obtained improvement in the VMA% of the HAMs was due to the reduction in their bulk specific gravity wherein the fibers could coat a wider surface area of the asphalt networks. In addition, the added fibers could absorb the binder, leading to an enhancement of the voids within the HAM networks. Additionally, at higher RWF contents, the compression was not uniform, thus producing higher AV% values. The obtained increase in the VMA% can be considered as an important feature for pavement construction in hot climates. This is mainly due to the bleeding tendency of the asphalt which, in turn, may increase void proportions and minimize the bleeding, thus facilitating more space for the binder to flow. The VMA values of the partial recycled aggregate samples were less than those contained for mixtures containing only natural aggregates. It was mainly due to the higher absorption of the bitumen with a porous surface due to RCAs. These findings are consistent with earlier observations [32,35].

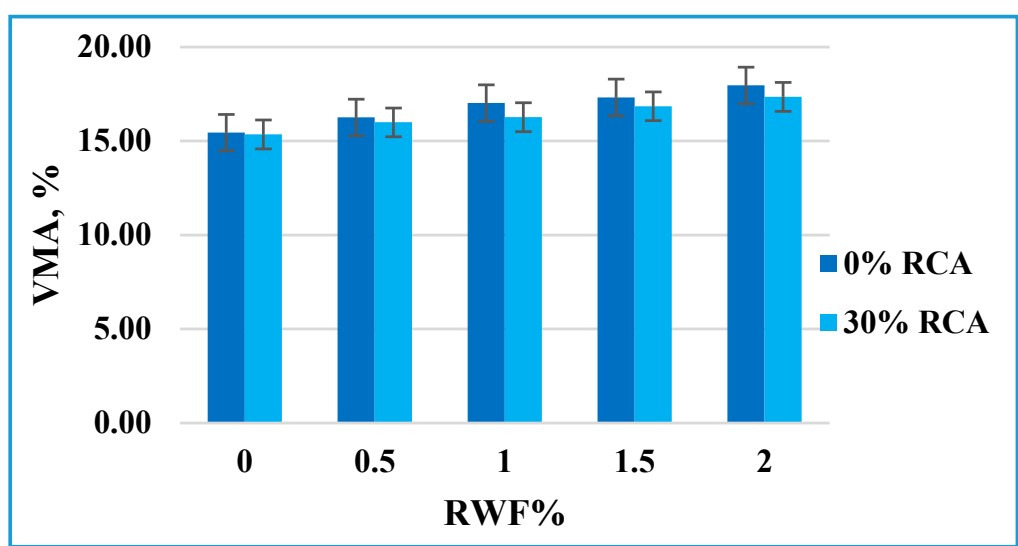

**Figure 11.** RWF content-dependent variation in the VMA of the mixes.

*3.2. Indirect Tensile Strength Ratio*

Table 3 illustrates the effect of different RWF-additive percentages on the TSR values of the HAMs obtained from ITS test. The achieved values of TSR were used to evaluate the mixtures' moisture susceptibility, differentiating it from the recommended moisture resistant capacity limit of 0.8. The control mix met the moisture susceptibility specifications. However, the mixes made with RWF showed an increase in the dry and wet ITS and TSR values until they reached a maximum at 1.5% RWF, then they began to decrease when the RWF content was increased to 2%. The dry ITS of the HAM was increased from about 9.67 to 23.31% when the corresponding RWF content was raised from 0.5 to 2% by weight of the total mixture (Figure 12). The wet ITS of the HAM was increased from 12.84 to 31.98%, while the TSR value was increased from 2.88 to 7.03% when the corresponding RWF content was raised from 0.5 to 2% (Figure 13). The dry and wet ITS values as well as TSR% of the HAM composed of 1.5% RWF were higher than those of other specimens. For the specimens made with 30% RCA, the ITS values for the unconditioned specimens first began to rise from 4.74 to 17.41 with an increase in RWF content from 0.5% to 1.5% by weight of the total mix. Then, the rate began to decrease by about −14.71%when the RWF content was 2%. For the conditioned specimens, the increase rates were 7.52, 24.33, and 2.85% for the RWF contents of 0.5, 1, and 1.5%, respectively. At 2% RWF content, the wet ITS value was decreased by about 15.98% compared to the sustainable control mix. The TSR value of the HAM increased and reached its highest level at a RWF content of 1%. Thereafter, the TSR value of the HAM dropped with the increase in the RWF level until it became below the accepted limit at 2% RWF. The rates of change were +2.65, +5.89, +2.34, and −1.49% for the RWF contents of 0.5, 1, 1.5, and 2%, respectively.

**Table 3.** Results obtained from the ITS test.

| RWF Additive (%) | 0% of RCA | | | 30% of RCA | | |
|---|---|---|---|---|---|---|
| | Dry ITS | Wet ITS | TSR (%) | Dry ITS | Wet ITS | TSR (%) |
| 0 | 1103 | 905.137 | 82.06 | 903.04 | 723.41 | 80.10 |
| 0.5 | 1209.7 | 1021.43 | 84.43 | 945.82 | 777.83 | 82.23 |
| 1 | 1340.5 | 1165.46 | 87 | 1060.23 | 899.43 | 84.83 |
| 1.5 | 1475.9 | 1349 | 91.39 | 907.54 | 744.05 | 81.98 |
| 2 | 1360.2 | 1194.6 | 87.83 | 770.17 | 607.76 | 78.91 |

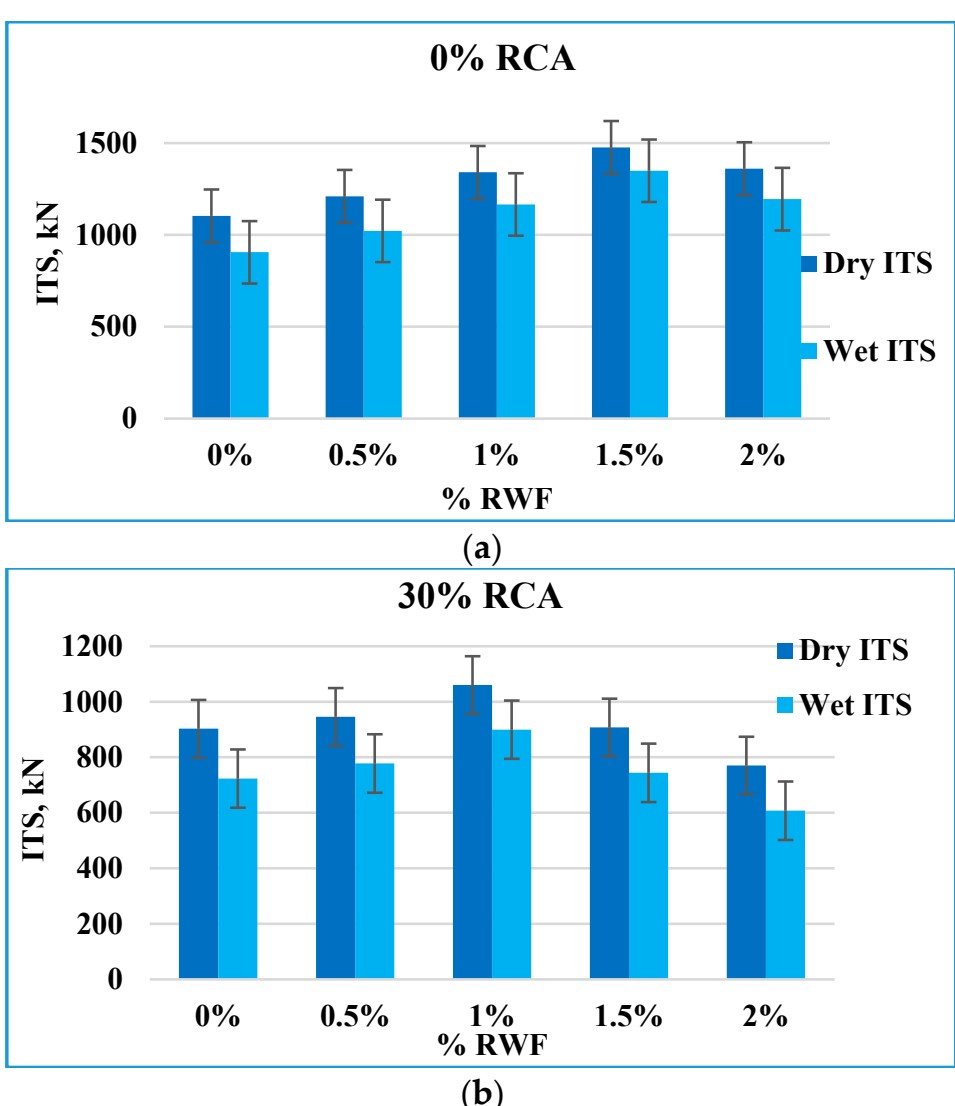

**Figure 12.** RWF content-dependent variation in the ITS of the mixes: (**a**) 0% (**b**) 30%.

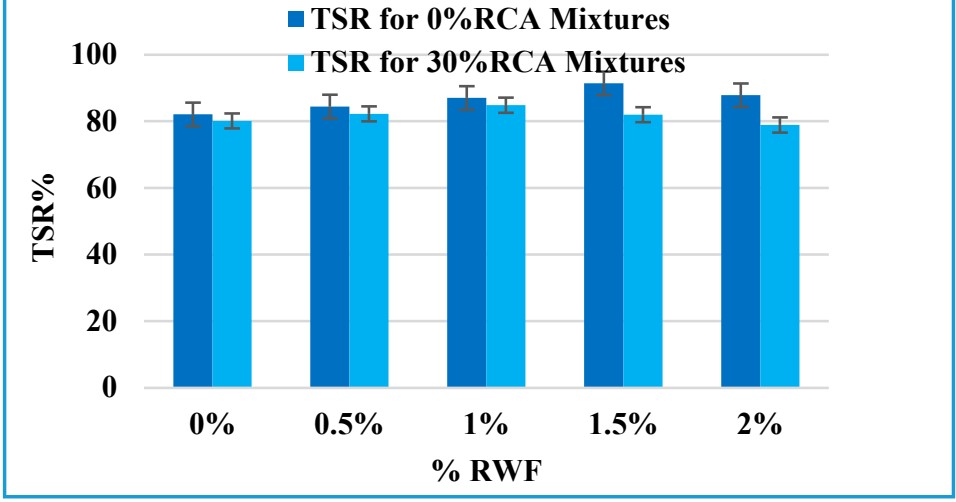

**Figure 13.** RWF content-dependent variation in the TSR of the HAM.

Dry and wet ITS values were higher for the mixtures containing fibers due to the interlocking among the constituents. This interlocking prevented the penetration of moisture into the mixture. The wet tensile strength was improved because the mixture became more resistant to damages caused by moisture. Briefly, RWF reinforcement of the HAMs played a considerable role in improving the tensile strength, making the specimen stiffer via the random orientation of the fibers within the HAM network structure. Consequently, this random dispersion of the fibers enabled a strong connection among the grains inside the network matrix, thereby preventing them from free movement. Small percentages of the fibers turned into fine material by heating and mixing, filling the voids between aggregate particles, thus preventing any water penetration. These results agreed with the report of Zhu, et al. [60] wherein basalt fibers were shown to improve the ITS values of both conditioned and unconditioned samples. In addition, the HAM made with 30% RCA demonstrated higher moisture sensitivity than the control mix that was further enhanced with RWF reinforcement. This improvement was mainly due to three reasons. First, the aggregates used in the concrete were round in shape, so when RCAs were crushed in the Los Angeles machine and treated with acids, most of the mortar was removed and the rounded shape of the aggregates was restored. It is well known that rounded aggregates do not provide desirable interlocking in asphalt mixtures, causing a weakness in the load transfer inside the mixes. Secondly, the residual mortar on the surface of the RCA made it more porous, and more water was absorbed because of the tendency of the cement to attract water. Another reason is that these residuals could prevent the perfect coverage of the asphalt binder on the aggregates, making stripping likely as was noticed during mixture preparation.

### 3.3. Index of Retained Strength

The obtained IRS values of the proposed HAMs were used to determine their damage resistance against moisture. According to SCRB [37], the minimal permissible IRS% value is 70%. Therefore, any mixture with an IRS% below this value can be regarded as being susceptible towards moisture damage. Table 4 illustrates the results of the IRS test. It was shown that the addition of RWFs into the HAMs as a partial replacement of RCAs could improve the CS values remarkably.

**Table 4.** Results obtained from the IRS tests of HAM.

| Additive (%) | 0% of RCA | | | 30% of RCA | | |
|---|---|---|---|---|---|---|
| | Dry CS (kPa) | Wet CS (kPa) | IRS (%) | Dry CS (kPa) | Wet CS (kPa) | IRS (%) |
| 0 | 6528 | 4980 | 76.28 | 6025 | 4382 | 72.73 |
| 0.5 | 7010 | 5728 | 81.71 | 6195 | 4850 | 78.28 |
| 1 | 7289 | 6254 | 85.80 | 6680 | 5344 | 80 |
| 1.5 | 8020 | 7045 | 87.84 | 6230 | 4620 | 74.15 |
| 2 | 7735 | 6045 | 78.15 | 5040 | 3420 | 67.85 |

Figure 14 shows the effect of various RWF contents on the CS values of the produced HAMs. The values of the dry CS, wet CS, and IRS of the designed HAMs were higher than those evaluated for the control mix. Compared to the control mix, the values of the dry CS of the HAMs were increased from 7.38 to 18.48% with the corresponding increase in the RWF content from 0.5 to 2%. In addition, the wet CS values for the HAMs were increased from 15.02 to 21.38% with the corresponding increase in the RWF content from 0.5 to 2%. Compared to the control mix, the values of IRS of the studied HAMs were enhanced by approximately 7.11, 12.47, 15.14, and 2.44% with corresponding increases in the RWF contents of 0.5, 1, 1.5, and 2%.

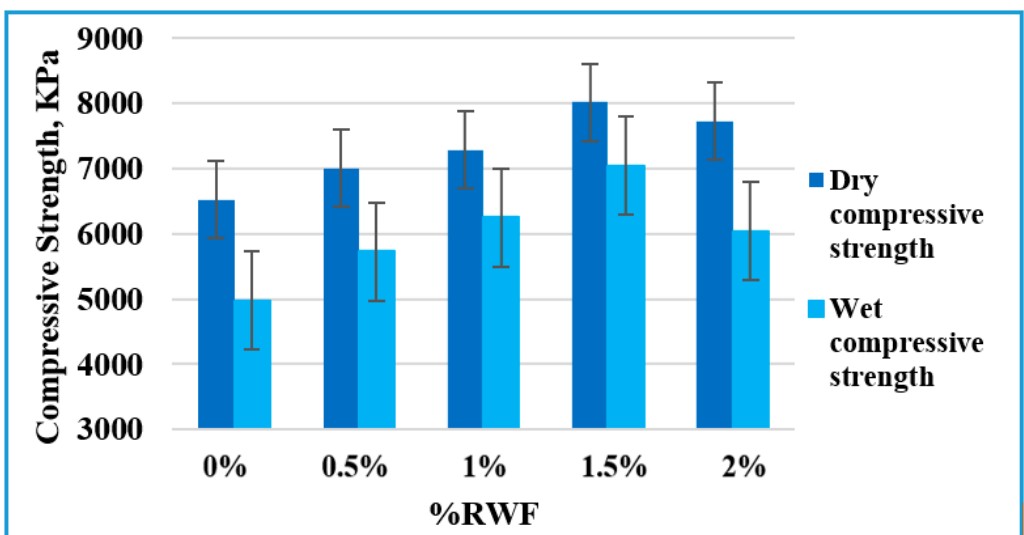

**Figure 14.** Effect of RWF contents on the CS values of HAMs.

Figure 15 shows the effect of various RWF contents on the IRS values of the proposed HAMs. The specimens prepared with 30% RCA showed a lower dry CS value than the control mix which was nearly 7.70 and 5.10% for RWF contents of 0 and 0.5%, respectively. After adding 1% RWF, the dry CS value of the specimen was increased by 2.32% compared to the control mix. Next, with an increase in fiber addition from 1.5 and 2%, the dry CS value of the HAMs dropped at a rate of 4.56 and 22.79, respectively, compared to the conventional control mix. The wet CS value of the HAMs was changed by about $-12$, $-2.61$, 7.30, $-7.22$, $-31.32$% for corresponding increases in the RWF contents of 0, 0.5, 1, 1.5, and 2% (Figure 16). The values of IRS were improved by 2.62 and 4.86% over the conventional control mix due to the inclusion of 0.5 and 1% RWF, respectively. However, the addition of 0, 1.5, and 2% RWF into the HAMs caused a decrease in the corresponding IRS value by about 4.66, 2.79, and 11.04% (Figure 17).

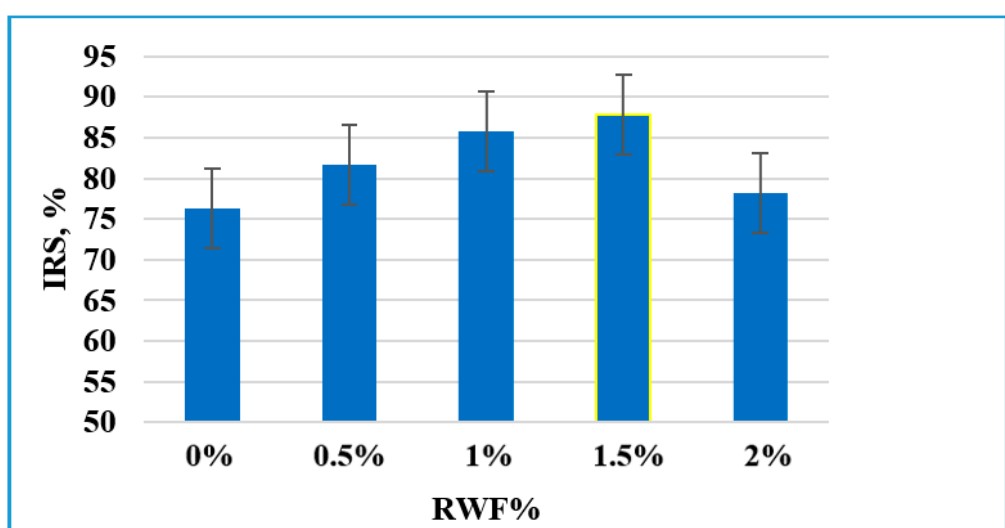

**Figure 15.** Effect of RWF contents on the IRS values of HAM.

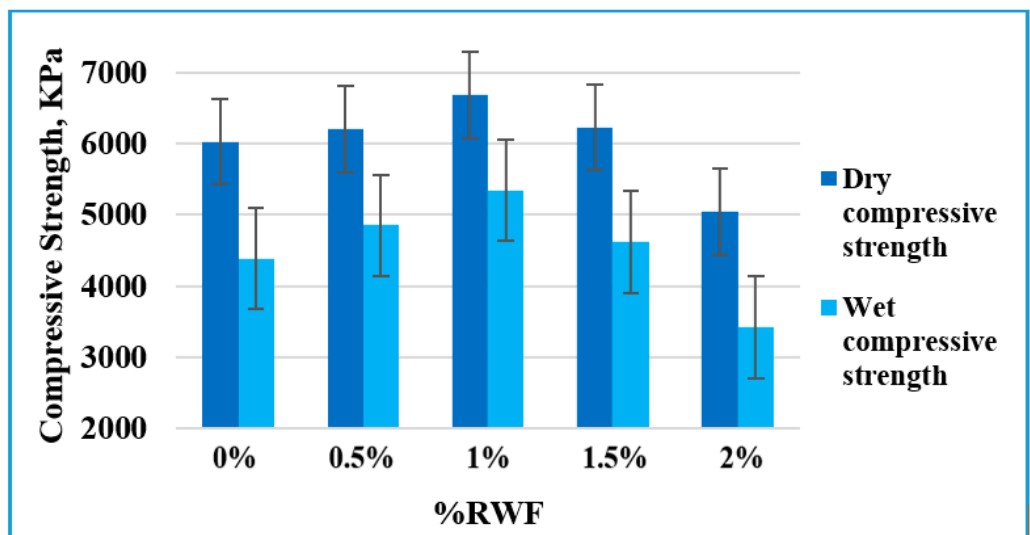

**Figure 16.** Effect of RWF on dry and wet CS values of HAM containing 30% RCA.

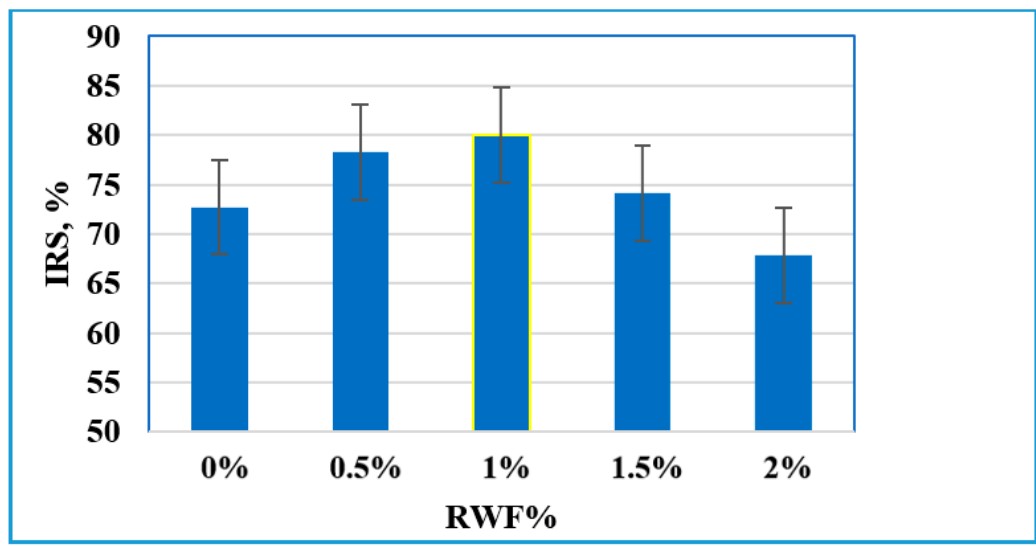

**Figure 17.** Effect of RWF on the IRS values of HMA containing RCA.

Figure 18 compares the RWF content-dependent IRS% of HAMs made without and with RCA. The current results are consistent with the observations of Mawat and Ismael [61] and Ali and Ismael [62] who showed that the addition of RWFs into HAMs can increase IRS values, thus improving moisture susceptibility. Compared to the control mix, the dry CS values of the sustainable HAMs containing 30% RCA was improved by about 2.82, 10.87, 3.40, and −16.34% with the 0.5, 1, 1.5, and 2% RWF reinforcement, respectively. Compared to the sustainable control mix, the wet CS values of the HAMs were increased by about 10.68, 21.95, and 5.43% for corresponding RWF contents of 0.5, 1, and 1.5% and then decreased by 21.95% when the RWF amount was raised to 2%. The IRS values were increased over the sustainable control mix by about 7.64, 10, and 1.96% with the addition of 0.5, 1, and 1.5% RWF and then dropped to about 6.70% when the RWF content was increased to 2%. From the obtained results, it can be claimed that the inclusion of RWF into the proposed HAMs could produce strong network bonds and prevent water penetration into the mix. This was the main reason for the higher increase in the wet CS values for the studied RWF-reinforced mixes. In addition, these mechanisms made the HAMs more durable and moisture-resistant against water-mediated damages. In brief, the incorporation of RWF into the HAMs was affirmed to be an innovative strategy in achieving excellent

moisture susceptibility desirable for high-performance pavement construction. It was acknowledged that [1], by replacing a part of the coarse aggregates with RCAs, the IRS values of the asphalt mixes can be enhanced appreciably. In contrast, our results disagreed with this claim and this may be due to the difference in the treatment process of RCAs.

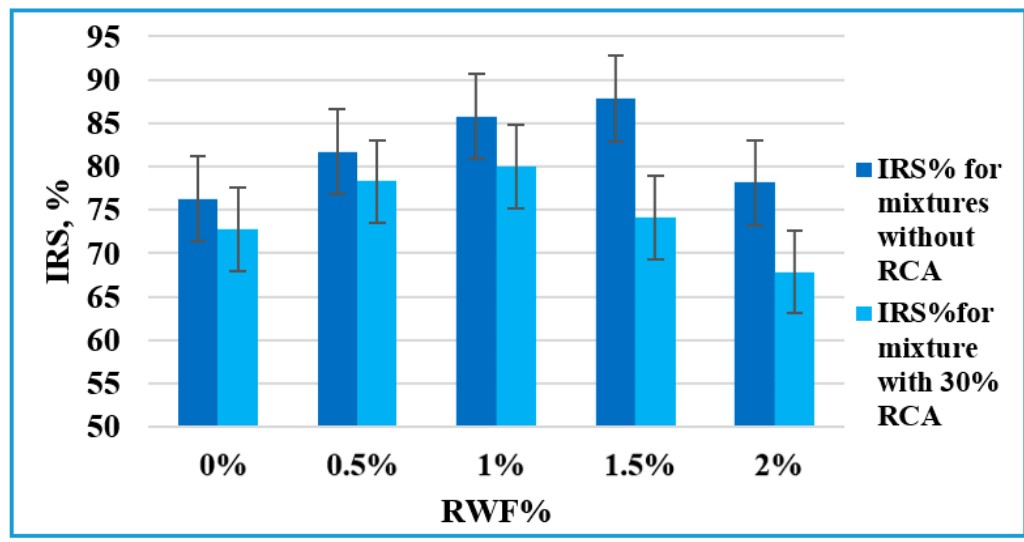

**Figure 18.** Effect of RWF on the IRS values of HAMs made without and with RCA.

Based on the results of ITS, TSR, CS, and IRS tests, it can be concluded that for the specimen made entirely using natural aggregates, the value of each indicator began to decrease with the increase in RWF content up to 2%. This fact can be attributed to a lack of bitumen coverage over the aggregates and fibers as well as to the difficulty of mixing when the time and heat of mixing were constant because of the large volume of fibers at this percent. All these effects led to less adhesion and allowed more water to enter into the mixture, thereby creating a more strippable mixture. Conversely, for the specimens made with the replacement of parts of their aggregates by RCAs, the above indicators values became lower than the optimum values at 1.5% RWF. This difference between these types of mixtures was mainly due to the lack of strong network bonding wherein the presence of RCA particles surrounded by cement mortar residuals were responsible for such weakening. Essentially, these two factors rendered the mixing process harder with less workability and a lack of bonding between mixture components. Thus, we observed decreases in the indicator values compared to those achieved for the first type of mixtures.

*3.4. FESEM Image Analysis*

Figures 19 and 20 display FESEM micrographs (magnifications between $50\times$ to $1000\times$) of the control mix and HAM prepared with 1.5% RWF reinforcement at, respectively. The HAM made with the RWF showed the formation of stable three-dimensional networks, enabling the development of an adhesive bituminous layer without allowing asphalt to seep down. This structural network can sustain severe temperature exposure by establishing grids of interconnected fibers among the aggregates and binder particles. RWFs can transfer and disperse forces from various directions when external stresses are applied to the asphalt mixture, minimizing the stress-mediated cracks in the HAM network. It is worth noting that RWFs have amorphous uneven morphology in addition to their usual cylindrical shape, wherein wider surface areas can absorb more asphalt particles and produce a thick coating of asphalt, thereby preventing moisture and water damage. The specimen also showed RWF roots that were tightly attached to the asphalt via interlocking, indicating that the RWF and asphalt had a high level of adhesion, effective for extreme moisture resistance and prevention of water penetration into the concrete.

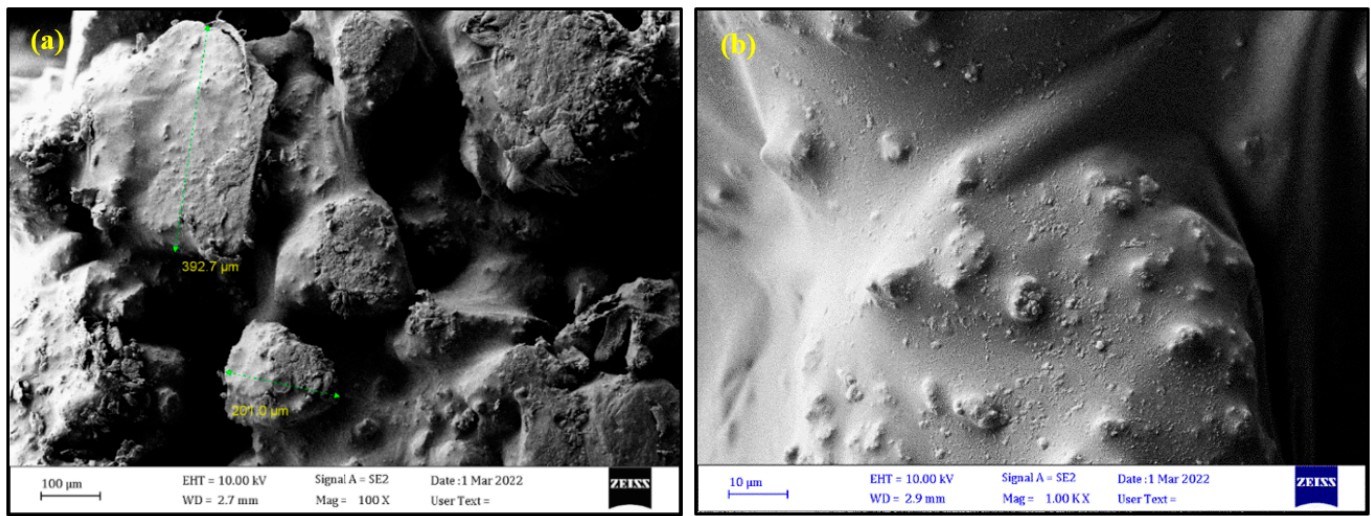

**Figure 19.** FESEM micrographs of the control mix with magnifications (**a**) 100×, (**b**) 1000×.

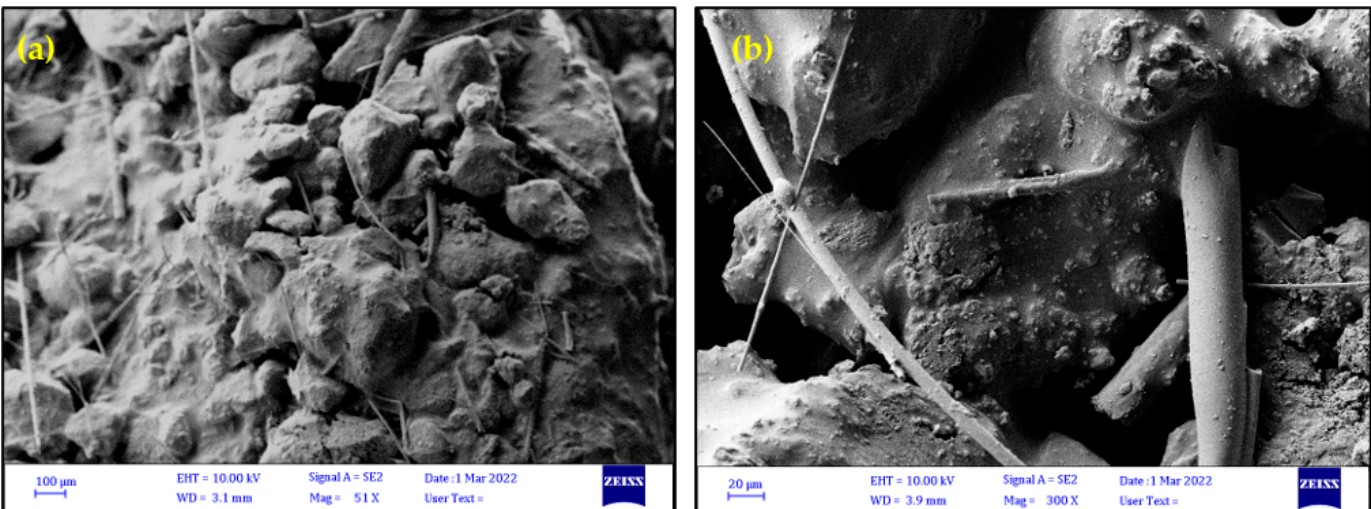

**Figure 20.** FESEM micrographs of the asphalt mixture reinforced with RWF with magnifications (**a**) 51×, (**b**) 300×.

## 4. Model Development for Prediction of Permanent Deformation of Modified Asphalt Concrete Pavement

Herein, the regression equation was used as a statistical prediction model. It represents the deterministic variation in the response variable, or dependent variable, according to other variables or independent variables, as well as the random component or error, which follows a specific probability distribution. The model was created using the SPSS program. This regression analysis aimed to produce adequate models using the available data at a chosen level of confidence while also fulfilling the fundamental propositions by offering the highest coefficient of determination (R2) and lowest standard error of estimate for the given data [63]. Various dependent and independent variables were used to simulate two permanent deformation models for the modified and conventional HAMs. A total of 15 samples for each model were used and the variables were TSR% (dependent), RWF% as additive (independent), and OAC% (independent).

### 4.1. Checking for Outliers

Chauvenet's criterion was used to identify influential observations and outliers. The outliers test was used to ensure data accuracy. Checking for outliers is important as it

enables one to recognize if any one of the observations is significantly different from the others [64]. Table 5 shows the complete absence of any outliers since the tabulated values are greater than the tests results.

**Table 5.** Results of Chauvenet's test for the outliers.

| Dependent Variable | Minimum ($X_{min}$) | Maximum ($X_{max}$) | Mean ($X'$) | SD (S) | $|(X_{min} - X')/S|$ | $|(X_{max} - X')/S|$ | $|(X_m - X')/S|$ |
|---|---|---|---|---|---|---|---|
| TSR | 80.99 | 91.77 | 86.54 | 3.311 | 1.676 | 1.579 | 2.13 |

*4.2. Stepwise Regression*

Stepwise regression analyses were performed to check whether the dependent variable was related to the variable's predictor so that the equation prediction could be calibrated. The final regression equation included all the available variables used in the statistical analyses. Table 6 presents the stepwise regression results.

**Table 6.** Summary of the stepwise regression analyses.

| Model | R | R Square | Adjusted R Square | Std. Error of Estimate |
|---|---|---|---|---|
| TSR | 0.979 | 0.958 | 0.951 | 0.73222 |

*4.3. Model Simulation Results*

Table 7 shows the standard error regression results for the permanent deformation in the proposed HAMs.

**Table 7.** Results of the final model.

| Stepwise Regression Model | $R^2$ | Adj. $R^2$ | SEE |
|---|---|---|---|
| TSR = 308.114 + 11.989 (RWF) − 46.067 (OAC) | 0.958 | 0.951 | 0.73222 |

**5. Conclusions**

In this study, RWF-reinforced and RCA-imbued HAMs were designed and their moisture susceptibility was determined when installed as highway pavements for the first time. Various tests were conducted to determine their CS, Marshall stability, air voids, flow, volumetric parameters and moisture sensitivity, moisture resistance, and microstructures. In addition, a model was developed to predict the permanent deformation in the modified asphalt concrete pavement due to moisture and water damage. Based on the results, the following conclusions were made:

i.   The HAM made with 1.5% RWF showed the highest CS and IRS values. Compared to the control mix, the HAM made with 1.5% RWF showed an optimum IRS and Marshall stability of 15.14% and 59.6%, respectively. This improvement was ascribed to the uniform distribution of the fibers into the hot asphalt network structure. Also, the flow values of the HAM dropped with increases in RWF contents, reaching a minimum at 2% RWF. The bulk density and VFA values of the HAM were decreased, and the AV% and VMA% were increased with an increase in RWF content. These results indicated that the proposed HAM is a potential candidate for pavement construction in hot climates;

ii.  The sensitivity of the HAM against moisture was decreased together with ITS (dry and wet) and TSR% until a certain level of RWF was reached. The maximum rate of change in the TSR% was 3.38% compared to the normal control mix and 5.89% compared to the sustainable control mix made with 1% RWF. The IRS, being an indicator of the moisture susceptibility of the HAM, showed an increase up to 4.86% with an increase in RWF content compared to the conventional control mix. Compared to the sustainable control mix, the IRS of the studied HAM was increased by 10% with addition of

1% RWF into the mix. In addition, the Marshall stability of the HAM was enhanced with the increase in RWF content, displaying a maximum of 39.39% compared to the conventional control mix and a maximum of 42.27% compared to the sustainable control mix containing 1% RWF by total weight of the mixture;

iii. The inclusion of 1% RWF into the hot mix asphalt with 30% of aggregate replacement by RCAs represented the optimum use of RWFs. The moisture susceptibility was decreased, thus meeting the requirements of Iraqi specifications regarding the Marshall stability, flow, AV%, and VMA%. The HAM made with 30% of virgin aggregates and RCAs revealed a decrease in the values of ITS, TSR, CS, Marshall stability, bulk density, VMA, and VFA. However, the OAC, AV%, and Marshall flow values of the HAM were increased;

iv. It was affirmed that proper recycling of demolished concrete into novel construction materials can reduce the amount of land required for waste disposal, reducing environmental pollution and global warming. In addition, recycled aggregates can minimize the energy expenditure needed to transport and produce aggregates, thus lowering greenhouse gases emissions. In places where the environment needs to be protected, this strategy can safeguard virgin aggregates and lands that serve as natural resources. Many European countries have enacted taxes on the use of virgin aggregates;

v. The model simulation supported the experimental observations. It is established that the proposed HAM with excellent moisture resistance can contribute to the development of construction sectors worldwide, thus meeting the requirements of sustainable goals.

**Author Contributions:** Conceptualization, F.K.H. and M.Q.I.; methodology, F.K.H.; software, F.K.H.; validation, F.K.H., M.Q.I. and G.F.H.; formal analysis, F.K.H.; investigation, M.Q.I.; resources, F.K.H.; data curation, F.K.H.; writing—original draft preparation, F.K.H.; writing—review and editing, G.F.H.; visualization, M.Q.I.; supervision, M.Q.I.; project administration, G.F.H.; funding acquisition, G.F.H. and M.Q.I. All authors have read and agreed to the published version of the manuscript.

**Funding:** This research received no external funding.

**Data Availability Statement:** Not applicable.

**Acknowledgments:** The authors thank the University of Baghdad for their support and cooperation in conducting this research.

**Conflicts of Interest:** The authors declare no conflict of interest.

## Abbreviations

| | |
|---|---|
| HAM | hot asphalt mixture |
| RCA | recycled concrete aggregate |
| RWF | rock wool fiber |
| OAC | optimal asphalt concentration |
| TSR | tensile strength ratio |

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
