# Peer review of "Rock Wool Fiber-Reinforced and Recycled Concrete Aggregate-Imbued Hot Asphalt Mixtures: Design and Moisture Susceptibility Evaluation"

_jcs, doi:10.3390/jcs7100428_

Round 1
Reviewer 1 Report
In this study, RWF-reinforced and RCA-imbued sustainable HAM are prepared to determine their moisture susceptibility when installed as highway pavements for the first time. Various tests are conducted to characterize their CS, Marshall stability, air voids, flow, volumetric parameters and the moisture sensitivity moisture resistance and micro-structures. It's well organized and structured. There are some issues needed to be clarified before it can be accepted for publication:
1. Currently, there have been many studies on recycled aggregates in building materials, and the author is suggested to add more newly published literature in recent years. e.g. The effect of steel fiber aspect-ratio and content on the fresh, flexural, and mechanical performance of concrete made with recycled fine aggregate. A review on the properties of recycled aggregate concrete (RAC) modified with nano-silica.
2. The description in the introduction is incoherent and has a weak logic.
3. Please adjust and optimize the format of the figure, e.g. Fig. 1 and Fig. 7.
4. Please revise typos and grammatical errors in the full manuscript. e.g. line 29-30.
5. The color of histogram in Fig. 15 needs to be consistent.
6. Please add the unit of compressive strength in Fig. 16.
7. The text in the SEM image (Fig .18) is not clear.
8. Please check the contents of Table 7.
9. Please add the error bar in the figure.
10. Please simplify the conclusions, 5 conclusions are applicable.
Good
Author Response
Reviewer' comments are highly appreciated. Please find the attached file (Response to reviewer' comments).

Reviewer 2 Report
[Major Comments]
This manuscript does not provide solid and insightful discussions. The authors consider whether the title and content are matched one more time.
The beginning of the introduction explains the situation in Iraq. So, this study seems like a case study in Iraq. So, Please introduce more cases all over the world.
So many literature reviews are not directly related to this study. The literature review concerning Rock wool fibers and Recycled concrete aggregates should be substantially revised.
The novelty of this study should be emphasized with convincing reasons from 196-206. This part seems the authors already knew the results. So, the authors did it. The research questions are required with problem statements.
Please provide why the mix design consists of five stages. An explanation is required for each stage.
The susceptibility in this study should be justified.
There are many narrative sentences in the results and discussions section. But discussions are not insightful at all.
Conclusions are too long with so many unimportant information. This section should be summarized with insightful comments.
[Minor Comments]
Lines 50-53: Please, add a reference.
Lines 56-57: Please, add a reference.
Lines 57-59: Please, add a reference.
Lines 66-68: Please, add a reference.
Lines 71-73: Please, add a reference.
Lines 97-99: Please, add a reference.
Lines 139-142: Please, add a reference.
Line 197: Avoid “we”
Line 245: Please, check the unit of temperature.
Line 282: Please, check the sentence, especially M.
Please, check all the equations.
Table designs must be satisfied with the journal regulations.
Please. check English by a native English speaker.
Author Response
Reviewer' comments are highly appreciated. Please find the attached file (Authors response to reviewers' comments).

Round 2
Reviewer 1 Report
The authors have revised the manuscript.
Author Response
We are really appreciated the reviewer' comment.

Reviewer 2 Report
This reviewer compares line by line between the original paper and the revised paper.
The authors re-arranged some parts and removed some parts.
There are no significant changes.
English is okay.
Author Response
Reviewer' comments are greatly appreciated. Following to reviewer' comment, we have amended and revised the manuscript. Please see the attached file.

Round 3
Reviewer 2 Report
Thank you for your great efforts.
Now, it is okay to publish in this journal.
Please, check English one more time. Just in case.